
# Using Two-Stream Theory to Capture Fluctuations of Satellite-Perceived TOA SW Radiances Reflected from Clouds over Ocean

Florian Tornow[1,2,3], Carlos Domenech[4], Howard W. Barker[5], René Preusker[1], and Jürgen Fischer[1]

[1]Institute for Space Sciences, Freie Universität Berlin, Berlin, Germany
[2]Earth Institute, Columbia University, New York, NY, USA
[3]NASA GISS, New York, NY, USA
[4]GMV, Madrid, Spain
[5]Environment and Climate Change Canada, Toronto, Ontario, Canada

**Correspondence:** florian.tornow@fu-berlin.de

**Abstract.** Shortwave (SW) fluxes estimated from broadband radiometry rely on empirically gathered and hemispherically resolved fields of outgoing top-of-atmosphere (TOA) radiances. This study aims to provide more accurate and precise fields of TOA SW radiances reflected from clouds over ocean by introducing a novel semi-physical model predicting radiances per narrow sun-observer geometry. Like previous approaches, this model was trained using CERES-measured radiances paired with

MODIS-retrieved cloud parameters as well as reanalysis-based geophysical parameters. By using radiative transfer approximations as a framework to ingest above parameters, the new approach incorporates cloud-top effective radius and above-cloud water vapor in addition to traditionally used cloud optical depth, cloud fraction, cloud phase, and surface wind speed. A two-stream cloud albedo—serving as a function of cloud optical thickness and cloud-top effective radius—and Cox-Munk ocean reflectance were used to describe an albedo over each CERES footprint. A simple equation of radiative transfer, with this

albedo and attenuating above-cloud water vapor as inputs, was used in its log-linear form to allow for statistical optimization. We identified the two-stream cloud albedo that minimized radiance residuals and outperformed the state-of-the-art approach for most observer-geometries and solar zenith angles between $20°$ and $70°$, reducing median standard deviations of radiance residuals per solar geometry by up to 13.2% for liquid clouds, 1.9% for ice clouds, and 35.8% for footprints containing both cloud phases. Tested for a variety of scenes, we further demonstrated the plausibility of scene-wise predicted radiance fields. This

new approach may prove useful when employed in Angular Distribution Models and may result in improved flux estimates, in particular dealing with clouds characterized by small or large droplet/crystal sizes.

## 1 Introduction

Radiative fluxes at top-of-atmosphere (TOA) inferred from satellite observations serve many purposes. Instantaneous flux estimates paired with properties of underlying clouds, aerosols, atmospheric gases, and Earth's surface may inform us about

the radiative effect of each component of Earth's radiation budget (e.g. Loeb and Manalo-Smith, 2005; Li et al., 2011; Thorsen et al., 2018). TOA fluxes may also help to constrain uncertainties concerning cloud-aerosol-radiation interactions, which will be



tested in the EarthCARE satellite mission (Illingworth et al., 2015). In EarthCARE's radiative closure assessment, observation-based fluxes will be used to help continuously assess both active-passive retrievals of cloud and aerosol properties, and results from radiative transfer simulations performed on them (Barker et al., 2011; Barker and Wehr, 2012). Integrals of estimated fluxes over large areas and long time spans (Loeb et al., 2018) help us understand the Earth-atmosphere system's current radiation budget (e.g. Stephens et al., 2012), thus helping to verify global climate models (e.g. Bender et al., 2006; Boucher et al., 2013; Calisto et al., 2014; Nam et al., 2012).

Inferring fluxes from satellite-based radiometry involves a number of steps. The key challenge for solar fluxes, the general focus of this paper, is that constituents such as clouds reflect solar radiation unevenly across the upward hemisphere and we need to assume how measurements from a subset of directions relate to radiances in directions not viewed. The intention is to adequately represent hemispheric distributions of radiances such that when integrated yield accurate flux estimates. The solution to this challenge has been empirical angular distribution models (ADMs) that learn, via statistical approaches, hemispherically-resolved radiance fields associated with atmospheric scenes using months of satellite observations. For clouds over ocean, the specific concern of this paper, early efforts (Suttles et al., 1988; Smith et al., 1986) worked with ERBE (Earth Radiation Budget Experiment) radiometry as well as GOES (Geostationary Operational Environmental Satellite) measurements and defined four scene types ranging in cloud coverage. Observations were sorted to produce mean radiances per observed angular ranges for each illumination geometry. Using CERES (Clouds and the Earth's Radiant Energy) and VIRS (Visible and Infrared Scanner) on the TRMM (Tropical Rainfall Measuring Mission) satellite, Loeb et al. (2003) refined this method and sorted observations into combinations of 12 cloud coverage classes and 14 cloud optical thickness groups and treated ice and liquid phase clouds separately. Instead of a discrete scene type definition, Loeb et al. (2005) defined a continuous description of scene type for the Terra mission, using a sigmoidal function to fit cloud optical thickness and cloud fraction based on MODIS (Moderate Resolution Imaging Spectroradiometer) measurements with CERES-measured TOA shortwave (SW) radiances. They treated footprints containing both ice and liquid phase clouds (throughout the paper referred to as "mixed phase") separately from pure and ice and liquid cases. Much of their state-of-the-art methodology was adapted for the Aqua mission (also hosting CERES and MODIS instruments) using improved cloud algorithms and longer data records (Su et al., 2015).

A recent case study (Tornow et al., 2018) focused on marine Stratocumulus-like clouds of optical thickness $\tilde{\tau} \approx 10$ and identified additional parameters that influence ADMs: above-cloud water vapor $ACWV$ and layer mean cloud-top effective radius $\overline{R_e}$. They showed that ignoring these parameters could cause deviations in instantaneous flux estimates of about 10 Wm$^{-2}$. This suggests the non-negligible role of single-scattering for determination of cloud reflectance patterns. Features of single-scattering, such as the cloud bow and glory for liquid clouds or the specular reflection peak for ice clouds, were generally visible in earlier ADMs (e.g. Loeb et al., 2005). These features – solely shaped by the particle phase function that largely depends on particle shape and size – can occur for a wide range of cloud optical thicknesses. Using simulated radiance fields, Gao et al. (2013) demonstrated that scattering regimes, ranging from foremost single-scattering to Lambertian-like multi-scattering mediums, are functions of the cloud optical thickness. For an intermediate regime, which showed single-scattering features, Gao et al. speculated that the uppermost $\tau \approx 1$ of cloud is responsible for single-scattering contributions.





This study presents a novel semi-statistical model that predicts TOA SW radiances for cloudy scenes over ocean for narrow ranges of Sun-observer angles. Estimates are sensitive to $\overline{R_e}$ and $ACWV$, and are compared to results from the state-of-the-art methodology. This new approach used the two-stream approximation to explain cloud albedo based on MODIS cloud properties and other geophysical auxiliary parameters. We began by finding the framework of approximations that best explained CERES-observed radiance fluctuations and then demonstrated that semi-physical log-linear models produced tenable radiance fields.

Section 2 presents data from Aqua and Terra satellites used in the current study. Section 3 explains both the state-of-the-art methodology for radiances estimation and the new approach. Section 4 identifies optimal solutions and assesses their properties. Section 5 discusses results and conclusions.

## 2 Data

Measured TOA SW radiances paired with scene properties – including imager-based cloud properties and further geophysical auxiliary parameters – were obtained from the CERES Ed4SSF (Edition 4.0 Single Scanner Footprint) dataset of Aqua and Terra satellite missions, primarily from days during years 2000-2005 when CERES instruments were measuring in rotating azimuth plane scan mode to provide angular coverage for ADM construction.

We extracted parameters concerning CERES broadband radiometry. Apart from upwelling unfiltered TOA SW radiances $I^*$, covering the spectral range of 0.4-4.5μm, and their angular geometry (i.e. solar zenith angle $\theta_0$, viewing zenith angle $\theta_v$, and relative azitmuth angle $\varphi$), we collected downwelling TOA SW fluxes $F^\downarrow$ that incorporate each measurements' prevalent Sun-Earth-distance which allowed normalization of gathered radiances via $I = I^* \frac{S_0 \cos\theta_0}{F^\downarrow}$, with solar constant $S_0 = 1361.0$ Wm$^{-2}$.

Collocated to each CERES footprint, the SSF dataset summarizes cloud property retrievals (Sun-Mack et al., 2018) on the MODIS pixel level taking into account the CERES point spread function (PSF) (Wielicki et al., 1996) and reports properties for up to two cloud layers per footprint (given that both layer's' cloud-top pressure differed by 50 hPa or more; Loeb et al., 2003). We extracted layer cloud fraction $f$, several statistics on the retrieved field of cloud optical depth $\tau$ (layer average of its logarithm $\tilde{\tau} = e^{\overline{\log\tau}}$, layer average $\overline{\tau}$, and layer standard deviation $\sigma(\tau)$), as well as layer mean values of cloud particle phase $\phi$ (involving MODIS band 3.7μm), effective radii of water or ice particles $\overline{R_e}$ (using band 3.7μm), and cloud-top pressure $p^{ctop}$. A quality flag summarizing the retrieval confidence ("Note for cloud layer") was also collected.

Additional geophysical auxiliary parameters provided in the SSF dataset were extracted. We obtained a surface broadband albedo $\alpha^{surface}$, surface IGBP (International Geosphere-Biosphere Programme) types, and 10 m surface wind speed $w_{10m}$. The wind speed parameter stemmed from GEOS data assimilation version 5.4.1.

Lastly, to incorporate above-cloud water vapor $ACWV$ into our analysis, we used layer mean cloud top pressure (of the layer with larger cloud fraction) and extracted from ERA-20C (ECMWF twentieth-century) reanalysis (Poli et al., 2016) four dimensional fields those vertical profiles of relative humidity $rh(p)$ and temperature $T(p)$ that were nearest in time and geolocation to the footprint center. For each CERES footprint we collocated the following vertical integral of mixing ratio





**Table 1.** Number of CERES footprints obtained after screening for marine clouds. Number are shown in million; in total 1 711 937 663 footprints.

| | No. of CERES footprints ($\times 10^6$) | | |
|---|---|---|---|
| Year | Terra | Aqua | Mode |
| | (FM1 & FM2) | (FM3 & FM4) | |
| 2000 | 164.02 | / | RAPS |
| 2001 | 228.10 | / | RAPS |
| 2002 | 236.74 | 84.39 | RAPS |
| 2003 | 236.60 | 203.30 | RAPS |
| 2004 | 243.53 | 245.65 | RAPS |
| 2005 | 6.05 | 63.56 | RAPS |

$mr(p)$, with saturation vapor pressure $e_s = 6.112 e^{\frac{17.67T}{T+243.5}}$ (using $T$ in degree Celsius) (Bolton, 1980) and molecular weights of water and dry air $mol_{h2o}$ and $mol_{air}$ respectively:

$$ACWV = \frac{1}{g} \int\limits_{p^{ctop}}^{0} mr(p,T,rh)dp = \frac{1}{g} \int\limits_{p^{ctop}}^{0} e_s(T) \frac{mol_{h20}}{mol_{air}} rh(p)dp \tag{1}$$

For our analysis, we filtered the extracted dataset for samples with more than 95% water surface, more than 0.1% cloud fraction, and solar zenith angles between 0° and 82°. Table 1 lists the resulting subset of 1.7 billion samples.

## 3 Methods for Capturing Radiance Fluctuations

In order to provide hemispherically resolved fields of backscattered radiances to radiance-to-flux-converting ADMs, statistical approaches capture observed radiances together with prevalent scene properties per narrow and discretized Sun-observer-geometry. Following Su et al. (2015), solar zenith, viewing zenith, relative azimuth angles were discretized into 2° intervals, referred to as $\theta_0^\Delta$, $\theta_v^\Delta$, and $\varphi^\Delta$, respectively. Combinations of $\theta_0^\Delta$, $\theta_v^\Delta$, and $\varphi^\Delta$ were denoted as angular bins and observations were sorted into bins for separate treatment. The following subsection presents the state-of-the-art methodology. Subsection 3.2 introduces a novel semi-physical approach that includes additional parameters.



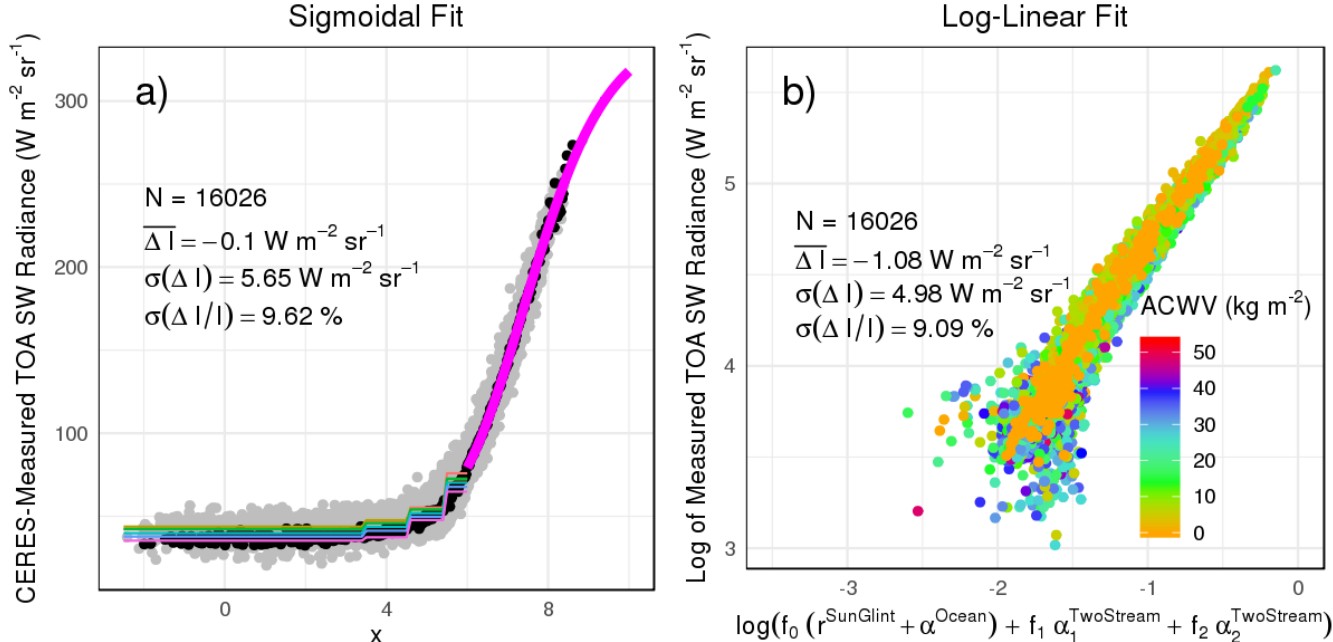

**Figure 1.** For an exemplary angular bin ($\theta_0 \in [20°, 22°]$, $\theta_v \in [6°, 8°]$, $\varphi \in [12°, 14°]$), we show how a state-of-the-art sigmoidal fit (a) and proposed log-linear model (b) capture fluctuations of CERES-measured TOA SW radiances. As this angular bin is within Sun-glint region, (a) shows the LUT-approach for $x < 6$ (as defined in Section 3.1; note that $f_1$ and $f_2$ are taken between 0-100 in (a) and between 0-1 in (b)). Colors in (b) mark the amount of above-cloud water vapor. Statistics in both panels summarize each approach's number of samples, bias, and standard deviation of radiance residuals as well as relative deviations.

## 3.1 State-of-the-Art Approach (Su et al., 2015)

An analytic sigmoidal function related TOA SW radiance with MODIS-based $f$ and $\tilde{\tau}$.

$$I(\theta_0^\Delta, \theta_v^\Delta, \varphi^\Delta) = I_0 + \frac{a}{[1 + e^{-\frac{(x-x_0)}{b}}]^c} \qquad (2)$$

Where $x = \log f\tilde{\tau}$ for a single cloud layer or $x = \log[(f_1 + f_2)e^{\frac{f_1 \log \tilde{\tau}_1 + f_2 \log \tilde{\tau}_2}{f_1 + f_2}}]$ for two layers and $I_0$, $a$, $b$, $c$, and $x_0$ were free parameters. Optimization of sigmoidal parameters relied on mean radiances that were produced per $x$ interval (every 0.02, shown as black dots in Figure 1a).

Models were generated separately per cloud phase. A footprint's cloud phase was determined via an effective phase, defined as $\phi_{eff} = \frac{f_1 \phi_1 + f_2 \phi_2}{f_1 + f_2}$ for two layers, and thresholds: liquid for $1 < \phi_{eff} < 1.01$, mixed for $1.01 \leq \phi_{eff} \leq 1.75$, and ice for $1.75 < \phi_{eff} \leq 2$.

To handle radiance fluctuations caused by sun-glint, a glint region was defined (sun glint angles $< 20°$). Observations with $x > 6$ in affected geometries remained captured by a sigmoid fit. For $x \leq 6$ on the other hand, a look-up-table approach stored





mean radiances per wind speed interval (0-2, 2-4, 4-6, 6-8, 8-10, and > 10 m s$^{-1}$) and per $x$ interval (<3.5, 3.5-4.5, 4.5-5.5, 5.5-6).

Selected angular bin in Fig. 1 had a sun-glint angle of about 14° and shows how tabulated radiances (colors correspond to wind speed intervals) and sigmoidal curve both covered observed radiances.

## 3.2 Novel Semi-Physical Approach

There are several ways one might incorporate additional variables $\overline{R_e}$ and $ACWV$ into a radiance-predicting statistical model. One could divide each angular bin's samples into classes of $\overline{R_e}$ and $ACWV$ and repeat sigmoidal fitting for each combination of classes (see Section 3.1). Some bins, however, contained too few samples or failed to cover the full spectrum of at least one of the two parameters. As a viable alternative, we explored radiative transfer approximations as a way to ingest scene properties (i.e. MODIS-based cloud properties and geophysical auxiliary parameters), and this allowed incorporating all samples in a continuous manner.

Working with cloudy atmospheres over ocean surfaces, we assumed that radiance fluctuations were mainly driven by the bidirectional reflection of clouds and water surfaces and by directional absorption through water vapor located above (highly reflective) clouds. We initially set out with following simple equation of radiative transfer:

$$I(\theta_0^\Delta, \theta_v^\Delta, \varphi^\Delta) \approx S_o \, \cos\theta_0 \, \alpha \, e^{-2ACWV} \tag{3}$$

with solar influx $S_o \cos\theta_0$ and the albedo $\alpha$ of an Earth-atmospheric scene covered by the CERES footprint (hereafter referred to as footprint albedo).

In following subsection, we present how footprint albedo was approximated. This then allowed us to use Eq. 3 in its log-linear form and weight the contribution of reflection and absorption via ordinary least square with free parameters $A$, $B$ and $C$.

$$\log I(\theta_0^\Delta, \theta_v^\Delta, \varphi^\Delta) \approx A + B \log\alpha + CACWV \tag{4}$$

Like the state-of-the-art methodology (Section 3.1), we applied this approach per angular bin (resolved by 2° in $\theta_0$, $\theta_v$, and $\varphi$) allowing us to treat $S_o \cos\theta_0$ as constant. We also separated by cloud phase but choose a different threshold to discriminate phase. As elaborated in more detail below, we rely on pure liquid and ice phases to, then, treat the mixed phase. Therefore, we consider a footprint as liquid phase for $\phi_{1/2} = 1$, as ice for $\phi_{1/2} = 2$, and as mixed for $\phi_1 = 1$ and $\phi_1 = 2$. $\phi_{1/2}$ were rounded in case their values were neither 1 or 2.

### 3.2.1 Approximating CERES Footprint Albedo

To approximate the albedo within each CERES footprint by means of MODIS-based cloud properties and additional geophysical variables ($\alpha^{surface}$, $w_{10m}$, $ACWV$), we separately handled clear and cloudy portions.

For clear portions within each footprint, we used the surface broadband albedo of underlying water bodies (referred to $\alpha^{ocean} = \alpha^{surface}$, see Section 2). To capture sun-glint, i.e. the specular reflection at ocean's surface that alters as low-level





winds perturb the water surface and tilt reflective facets, we used a Cox-Munk reflectance (Cox and Munk, 1954), as formulated in Wald and Monget (1983) with Fresnel reflection factor $\rho(\omega)$ for a perfectly smooth surface, and sun-observer-geometry per CERES footprint:

$$r^{SunGlint} = \frac{\pi \rho(\omega) P(\theta_n, W_{10m})}{4 \cos \theta_0 \cos \theta_v \cos^4 \theta_n} \tag{5}$$

where

$$P(\theta_n, W_{10m}) = \frac{1}{\pi \sigma^2} \exp\left(-\frac{\tan^2 \theta_n}{\sigma^2}\right) \tag{6}$$

$$\sigma^2 = 0.003 + 0.00512 W_{10m} \tag{7}$$

$$\theta_n = \arccos\left(\frac{\cos \theta_v + \cos \theta_0}{2 \cos \omega}\right) \tag{8}$$

$$\cos 2\omega = \cos \theta_v \cos \theta_0 + \sin \theta_v \sin \theta_0 \cos \varphi \tag{9}$$

To describe the albedo of cloudy portions, we explored the application of two-stream cloud albedo that uses cloud optical thickness and cloud micro-physical properties through asymmetry parameter $g$ or backscattering fraction $\beta$. This allowed us to ingest MODIS-based $\tilde{\tau}$ and $\overline{R_e}$ through $g(\overline{R_e})$, as explained in more detail in the following subsection. The following solutions are thoroughly described in Meador and Weaver (1980), which presents a unifying theoretical framework to a variety of two-stream cloud albedos based on coupled differential equations that describe upward and downward directed intensity fields. We considered two cloud albedos that proved useful for a range of cloud optical thicknesses (King and Harshvardhan, 1986): the Eddington approximation (Shettle and Weinman, 1970) and the Coakley-Chylek approximation (using solution I of Coakley and Chylek, 1975).

The Eddington approximation considered an incident flux explicitly in coupled equations and thus described diffuse intensity fields. Assuming conservative scattering (i.e. a single scattering albedo of 1), a perfectly absorbing lower boundary ($\alpha^{bottom} = 0$), and no further influx at TOA, the analytical solution for cloud albedo was as follows, where $\mu_0 = \cos \theta_0$:

$$\alpha^{TwoStream} = \frac{\frac{3}{4}(1-g)\tilde{\tau} - \frac{1}{4}(1-3\mu_0)(1 - e^{-\frac{\tilde{\tau}}{\mu_0}})}{1 + \frac{3}{4}(1-g)\tilde{\tau}} \tag{10}$$

The Coakley-Chylek approximation excluded the incident flux in differential equations and thus its intensities referred to total radiation fields (i.e. direct and diffuse). Assuming conservative scattering, a perfectly absorbing lower boundary ($\alpha^{bottom} = 0$) and only a solar influx at TOA, the analytical solution for cloud albedo was:

$$\alpha^{TwoStream} = \frac{\frac{(1-g)\tilde{\tau}}{2}}{1 + \frac{(1-g)\tilde{\tau}}{2}} \tag{11}$$





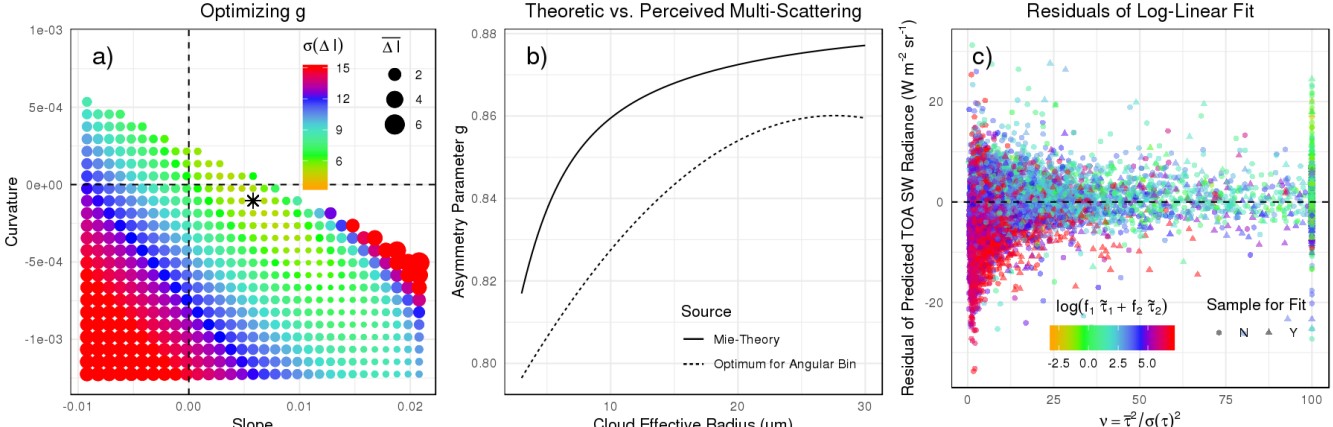

**Figure 2.** For the same angular bin as in Figure 1, we present details of the proposed model that highlight essential steps aside from log-linear least-square fitting (Eq. 4). (a) shows the search for an optimal $g(\overline{R_e})$ (as described in Sec. 3.2): we plotted a two-dimensional slice (showing $b$ and $c$ of Eq. 14) through the three-dimensional space (spanned by $a$, $b$, and $c$). Colors show standard deviations of radiance residuals and point size relates to model bias. The star marks the combination of $a$, $b$, and $c$ that produced smallest residual standard deviations and is considered optimal for this bin. (b) compares the $g(\overline{R_e})$ of the determined optimal solution against Mie-calculations. (c) shows final radiance residuals against cloud homogeneity (x-axis) and cloud optical depth (color). As described in Section 3.2.2., only homogeneous ($\nu > 10$) clouds which were well-retrieved (MODIS-reported portion $> 90\%$) - marked as triangles in (c) - were considered for opimization of $g(\overline{R_e})$ and least-square fitting. Statistcs and error metrics throughout the manuscript incorporate all samples.

where $\beta$ was substituted with $\frac{(1-g)}{2}$ as done in textbook solutions (e.g. Bohren and Clothiaux, 2008). Using the Coakley-Chylek approximation and a reflective lower boundary with albedo $\alpha^{bottom} > 0$ (in this study $\alpha^{bottom} = \alpha^{surface}$), we produced following cloud albedo:

$$\alpha^{TwoStream} = \frac{\alpha^{ocean} + \frac{(1-\alpha^{ocean})(1-g)\tilde{\tau}}{2}}{1 + \frac{(1-\alpha^{ocean})(1-g)\tilde{\tau}}{2}} \tag{12}$$

Because it was unclear which solution could explain radiance fluctuations over narrow sun-observer-geometries most suc-
cessfully, we tested a variety of solutions in Section 4.

Both ocean and cloud albedos (for up to two cloud layers) were used to calculate the footprint albedo, using clear fraction $f_0$ and cloud fractions of layer 1 and layer 2, $f_1$ and $f_2$, respectively:

$$\alpha = f_0(\alpha^{ocean} + r^{SunGlint}) + f_1\alpha_1^{TwoStream} + f_2\alpha_2^{TwoStream} \tag{13}$$

where $f_0 + f_1 + f_2 = 1$.

**3.2.2  Statistical Optimization**

Before comparing different two-stream approximations in Sec. 4, we performed two steps that ensured statistical optimization for each approximation. Finding an optimal $g(\overline{R_e})$ was designed to best capture radiance fluctuations per angular bin. Higher





**Table 2.** In search for optimal $g(\overline{R_e})$, we list the range (Minimum and Maximum) and step size for each parameter in Eq. 14.

| Parameter | Minimum | Maximum | Step |
|---|---|---|---|
| a | -0.5 | 0.95 | 0.01 |
| b | -0.01 | 0.01 | 0.0003 |
| c | -0.00025 | 0.00025 | 0.000015 |

weights to a subset of data per angular bin - homogeneous clouds that were well retrieved - was used to facilitate consistency of radiances across bins. Both steps are explained in more detail below.

As shown in the previous subsection, we used two-stream cloud albedo to explain radiance fluctuations for narrow sun-observer-geometries. Applied to all angular bins of an upward hemisphere, it was unclear which $g(\overline{R_e})$ to choose. Initial tests that used a $g(\overline{R_e})$ from Mie theory (see Fig. 2b) for all geometries proved sub-optimal for some angular bins and left radiance residuals correlated to layer mean effective radius (not shown). We therefore decided to optimize $g(\overline{R_e})$ for each angular bin and for each cloud phase (liquid and ice). Inspired by the shape of Mie-calculated $g(\overline{R_e})$, we approximated$g(\overline{R_e})$ via a

quadratic function:

$$g(\overline{R_e}) = a + b\overline{R_e} + c\overline{R_e}^2 \tag{14}$$

and searched a three-dimensional grid, spanned by $a$, $b$, and $c$, for combinations that minimized the standard deviation of radiance residuals. The search covered parameters $a$, $b$, and $c$ as listed in Table 2. As shown in Fig. 2a, we usually found a single optimum value that could minimize standard deviation of radiance residuals and that deviated from Mie calculations

(Fig. 2b).

A second step aimed at using a subset of data that was consistent across angular bins. Looking into samples of individual angular bins, we observed stark variability in radiances that could be attributed to cloud horizontal heterogeneity (cloud homogeneity was approximated by $\nu = \frac{\bar{\tau}^2}{\sigma(\tau)^2}$; radiance residuals are shown in Fig. 2c). We suspected that clouds' three-dimensional structure caused tilted cloud facets that led to more or less reflective cloud portions (e.g. as accounted for in Scheck et al.,

2018). In order to avoid an uncontrollable impact of cloud heterogeneity onto final models, we decided to select homogeneous samples only for statistical optimization. As a threshold of homogeneity, we used $\nu > 10$ (e.g. Barker et al., 1996; Kato et al., 2005). As shown in Tab. 3 per solar geometry, median homogeneity varied considerably across bins as well as cloud phase, and this resulted in ranging portions of data being selected. For optimization, we further limited selection to CERES footprints with quality flags indicating a confident retrieval of 80% or more of all cloudy MODIS pixels within a CERES footprint. This

subset of samples served to optimize the above search for $g(\overline{R_e})$ and to find weights via least-square (Eq. 4). To compute error metrics, we used all available samples. An example for the application of the log-linear model in shown in Fig. 1b.





**Table 3.** Per solar geometry $\theta_0$ and per cloud phase (L - liquid, I - ice, M - mixed) as defined in Section 3.2, we show what portions of the upward hemisphere were covered with observations, and how large the range of cloud homogeneity, above-cloud water vapor, cloud-top effective radius was. The range lists minima and maxima of median values computed per angular bin within a hemisphere.

| $\theta_0^{\Delta}$ (in °) | Angular Coverage | | | $\nu$ | | $ACWV$ (kg m$^{-2}$) | | $\overline{R_e}$ ($\mu$m) | |
|---|---|---|---|---|---|---|---|---|---|
| | L | I | M | L | I | L | I | L | I |
| 6-8 | 0.13 | 0.01 | 0.16 | 2.3-4.4 | 5.2-12.2 | 11.10-16.52 | 0.03-0.04 | 11.4-14.5 | 39.8-45.7 |
| 8-10 | 0.23 | 0.05 | 0.25 | 1.7-4.8 | 4.3-10.4 | 9.29-16.85 | 0.02-0.05 | 11.5-13.5 | 38.6-47.8 |
| 10-12 | 0.35 | 0.12 | 0.35 | 2.3-10.9 | 3.7-15.8 | 9.39-17.37 | 0.02-0.06 | 10.2-13.3 | 38.5-49.6 |
| 12-14 | 0.59 | 0.20 | 0.58 | 2.4-9.4 | 3.0-17.1 | 7.76-16.79 | 0.02-0.06 | 10.1-15.2 | 38.7-47.5 |
| 14-16 | 0.75 | 0.34 | 0.77 | 2.3-7.5 | 3.2-22.4 | 6.31-18.78 | 0.02-0.08 | 9.9-18.3 | 36.7-74.3 |
| 16-18 | 0.83 | 0.61 | 0.85 | 1.8-7.2 | 3.4-24.9 | 7.17-21.15 | 0.02-0.06 | 9.6-20.5 | 36.3-47.3 |
| 18-20 | 0.88 | 0.71 | 0.89 | 2.0-8.5 | 3.1-20.2 | 7.68-19.19 | 0.02-0.07 | 9.5-20.0 | 37.4-47.2 |
| 20-22 | 0.91 | 0.76 | 0.93 | 2.0-9.1 | 3.0-16.2 | 6.77-19.97 | 0.02-0.06 | 10.0-19.5 | 39.1-49.9 |
| 22-24 | 0.93 | 0.79 | 0.95 | 2.0-9.1 | 3.5-16.1 | 7.00-18.79 | 0.02-0.06 | 10.0-19.4 | 38.6-49.0 |
| 24-26 | 0.94 | 0.81 | 0.96 | 2.1-8.0 | 3.6-15.5 | 7.26-18.18 | 0.02-0.07 | 9.8-18.8 | 40.1-61.0 |
| 26-28 | 0.94 | 0.82 | 0.97 | 1.9-8.3 | 3.2-15.5 | 7.23-16.98 | 0.02-0.06 | 10.3-16.9 | 40.2-51.9 |
| 28-30 | 0.95 | 0.83 | 0.97 | 2.0-11.0 | 3.3-15.8 | 6.72-20.07 | 0.02-0.07 | 10.8-16.2 | 39.5-51.6 |
| 30-32 | 0.95 | 0.83 | 0.98 | 1.9-10.0 | 3.4-16.4 | 7.04-17.02 | 0.02-0.08 | 11.2-17.2 | 40.7-73.2 |
| 32-34 | 0.95 | 0.83 | 0.98 | 2.0-8.8 | 3.1-17.0 | 7.06-17.99 | 0.02-0.10 | 11.6-16.9 | 41.0-55.5 |
| 34-36 | 0.96 | 0.83 | 0.98 | 2.0-8.8 | 3.5-17.2 | 6.44-18.76 | 0.02-0.11 | 11.8-16.0 | 41.9-52.8 |
| 36-38 | 0.96 | 0.83 | 0.98 | 1.9-6.9 | 3.9-18.8 | 6.66-18.18 | 0.02-0.13 | 11.9-15.8 | 42.2-56.6 |
| 38-40 | 0.95 | 0.83 | 0.98 | 2.0-7.7 | 4.4-20.0 | 6.61-16.88 | 0.02-0.13 | 11.7-15.8 | 42.2-54.8 |
| 40-42 | 0.95 | 0.83 | 0.98 | 1.9-7.7 | 5.4-20.6 | 5.55-19.06 | 0.03-0.14 | 11.5-16.3 | 43.6-55.8 |
| 42-44 | 0.94 | 0.82 | 0.98 | 2.0-7.1 | 5.0-20.3 | 5.94-15.28 | 0.03-0.16 | 11.6-16.5 | 43.0-62.8 |
| 44-46 | 0.94 | 0.83 | 0.98 | 2.1-8.1 | 5.7-20.5 | 5.90-12.69 | 0.04-0.15 | 11.5-16.4 | 43.7-74.4 |
| 46-48 | 0.94 | 0.83 | 0.97 | 2.1-7.9 | 6.0-19.4 | 5.37-12.45 | 0.03-0.15 | 11.6-15.6 | 44.3-78.2 |
| 48-50 | 0.93 | 0.83 | 0.97 | 2.0-7.3 | 6.1-17.9 | 4.97-12.23 | 0.04-0.16 | 11.7-15.0 | 45.3-76.1 |
| 50-52 | 0.92 | 0.83 | 0.97 | 2.0-7.3 | 5.7-18.1 | 4.55-10.34 | 0.05-0.16 | 11.6-14.8 | 44.6-80.0 |
| 52-54 | 0.91 | 0.83 | 0.96 | 2.2-7.5 | 5.6-16.6 | 4.09-9.76 | 0.00-0.17 | 11.8-15.0 | 46.1-77.0 |
| 54-56 | 0.90 | 0.83 | 0.96 | 2.1-7.0 | 5.5-14.9 | 3.88-8.67 | 0.04-0.16 | 11.0-15.7 | 46.1-78.6 |
| 56-58 | 0.89 | 0.83 | 0.96 | 3.0-7.4 | 5.2-13.3 | 3.30-8.88 | 0.05-0.15 | 10.8-15.0 | 45.0-76.2 |
| 58-60 | 0.88 | 0.83 | 0.96 | 3.1-8.8 | 4.6-12.4 | 2.99-8.77 | 0.05-0.15 | 10.7-15.1 | 44.0-76.9 |
| 60-62 | 0.87 | 0.84 | 0.95 | 2.8-8.7 | 4.8-11.7 | 3.24-10.05 | 0.05-0.14 | 11.2-15.2 | 43.8-76.8 |
| 62-64 | 0.86 | 0.83 | 0.95 | 3.4-8.7 | 4.5-13.3 | 2.74-7.62 | 0.04-0.15 | 11.5-14.9 | 44.1-76.2 |
| 64-66 | 0.85 | 0.84 | 0.94 | 3.5-9.6 | 3.9-11.7 | 2.32-7.73 | 0.04-0.15 | 11.5-14.7 | 42.9-79.5 |
| 66-68 | 0.85 | 0.84 | 0.94 | 3.5-9.3 | 3.5-10.6 | 2.37-8.56 | 0.05-0.15 | 11.6-15.2 | 44.1-82.2 |
| 68-70 | 0.84 | 0.84 | 0.93 | 3.8-11.2 | 3.2-10.1 | 2.30-9.00 | 0.05-0.15 | 11.5-15.3 | 41.0-75.6 |
| 70-72 | 0.83 | 0.83 | 0.93 | 3.9-11.6 | 2.7-10.2 | 2.45-7.02 | 0.05-0.16 | 11.7-15.1 | 42.3-76.1 |
| 72-74 | 0.82 | 0.82 | 0.92 | 3.8-12.7 | 2.2-9.1 | 2.05-6.44 | 0.06-0.15 | 11.8-15.8 | 40.6-86.2 |
| 74-76 | 0.80 | 0.81 | 0.91 | 4.0-10.9 | 1.8-10.0 | 1.98-7.46 | 0.04-0.15 | 11.1-16.1 | 40.0-76.6 |
| 76-78 | 0.79 | 0.79 | 0.90 | 3.8-13.0 | 1.5-8.8 | 1.87-9.36 | 0.06-0.15 | 11.6-15.8 | 39.2-88.6 |



## 4 Results

Radiance-predicting statistical models that capture narrow sun-observer-geometries form the basis for empirical angular distribution models. And these statistical models fit observations from satellites, typically capturing how TOA SW radiances

measured by a broadband radiometer change with scene type (defined by surface conditions as well as cloud and aerosol properties within the radiometer's footprint area) retrieved using a multi-spectral imager (see Sec. 2). To investigate whether a new approach, the proposed semi-physical log-linear model in Sec. 3.2, is a superior way to fit observations compared to the state-of-the-art approach, the sigmoidal fit described in Sec. 3.1, we took CERES Ed4SSF observations (Sec. 2) of liquid-phase clouds along the principal plane of an exemplary solar geometry covering major scattering features of clouds and the

ocean surface. We applied the sigmoidal fit as well as a variety of log-linear models, each using a different analytic solution of two-stream cloud albedo (Eqs. 10-12) that is used in this study as a framework to ingest MODIS-based cloud properties. Looking at the standard deviation of radiance residuals per angular bin (in this study used as a measure of model uncertainty), the Coakley-Chylek approximation using a reflective lower boundary (Eq. 12) outperformed the sigmoidal fit for most bins and by up to 1.5 W m$^{-2}$ sr$^{-1}$ (shown in Fig. 3). Only the central portion of sun-glint-affected geometries remained best ex-

plained by sigmoidal fits (and accompanied look-up-table approach as laid out in Section 3.1). In contrast, the Coakley-Chylek approximation using a perfectly absorbing lower boundary (Eq. 11) or the Eddingtion approximation (Eq. 10) performed only equally well or worse than sigmoidal fits.

To ensure that the Coakley-Chylek approximation using a reflective lower boundary performed well for other Sun-observer-geometries, we processed all angular bins that contained more than 100 samples. As Table 3 shows, this covered between 13

and 96% of all angular bins. We found that for liquid clouds (top panels of Fig. 4) and $\theta_0 \sim 20°$-$70°$ more than half of the bins were better explained by the log-linear approach and errors were reduced by up to 13.2%. For solar geometries $\theta_0 > 40°$, bins in sun-glint-affected geometries (constituting a portion of all bins in a hemisphere between 10% for a $\theta_0 \sim 20°$ and 1% for a $\theta_0 \sim 75°$) caused higher uncertainties in log-linear models. For solar geometries $\theta_0 < 20°$, on the other hand, we found bins outside the sun-glint – i.e. mostly slant observation angles – were best treated with the sigmoidal approach. Few footprints

(indicated by circle size) of the top row were treated as mixed in the log-linear model and will be evaluated further below. With these limitations in mind, we use the Coakley-Chylek approximation using a reflective lower boundary standard as two-stream cloud albedo for the remainder of this study.

To determine whether the log-linear approach predicted plausible radiance fields, we tested it on a variety of scenarios. When applied to a range of cloud optical thicknesses, we found a similar radiance response compared to sigmoidal fit (Fig. 5b). Setting

cloud fraction to zero ($f_1 = f_2 = 0$) and using a range of 10 m wind speeds, log-linear and sigmoidal models produced again comparable radiance fields (Fig. 5c). This shows that the sensitivities of the state-of-the-art approach were captured by log-linear models. When varying cloud-top effective radius – a newly added sensitivity – we found radiances grow as droplet size increased (leaving cloud optical thickness constant; shown in Fig. 5e). With a focus on single-scattering features, we found the cloud glory (centered around the direct backscatter) to widen and the cloud glory (positioned about $20°$ away from

the backscatter) to shift towards the direct backscatter as effective radii became smaller. This observation is corroborated by

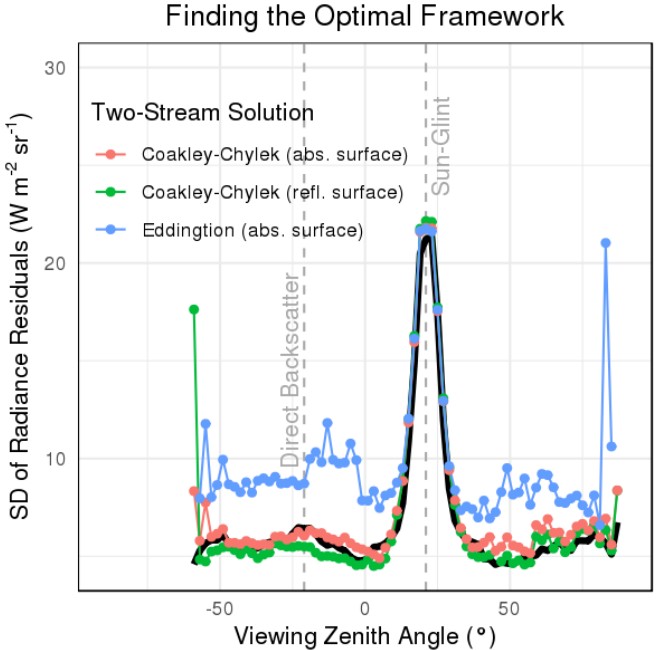

**Figure 3.** Applied to angular bins of the principal plane for $\theta_0 \in [20°, 22°]$, we test a variety of two-stream solutions for cloud albedo (Eqs. 10-12) as input to log-linear models as presented in Eq. 4. This plot shows standard deviations of resulting residuals and compares against the state-of-the-art sigmoidal fit (black line). As labelled, grey dashed lines mark position of sun-glint and direct backscatter.

Mie-calculations of scattering phase functions (e.g. Fig. 1 in Tornow et al., 2018). The newly introduced concept of bin-wise optimized asymmetry parameters (Sec. 3.2.2) made changing cloud bow and glory possible and $g(\overline{R_e})$ exhibited a symmetry left and right of the direct backscatter between $\theta_v$ of -50° and 0° (Fig. 5f). For a range of above-cloud water vapor (Fig. 5d) – another newly added sensitivity - we observed that smaller loads produced higher radiances and found a slight increase in

sensitivity with larger $\theta_v$ .

We also tested log-linear models on observations of ice-phase clouds. We found that model uncertainties outside the sun glint were of similar magnitude as sigmoidal fits (Fig. 4, bottom panels). Possible reasons will be discussed in Sec. 5. Like the liquid-phase, predicted radiances increased with smaller ice crystal radii. However, distinct scattering features were absent (not shown); possibly a result of ice clouds' rich variety of crystal shapes (e.g. Zhang et al., 1999; Baum et al., 2005) that was

unaccounted for. The response to above-cloud water vapor was consistent and covered much of the lower levels (0.03-0.17 kg m$^{-2}$, see Table 3).

Roughly 50% of all CERES footprints cover both a liquid and an ice cloud and have been treated separately as "mixed-phase". The proposed log-linear approach allows us to handle mixed-phase cases fundamentally differently. Instead of a footprint-effective optical depth (as used in Equ. 2), we can produce a footprint-effective albedo (Equ. 13) and account not

only for cloud macro-physical ($f_{1/2}, \tilde{\tau}_{1/2}$) but also for microphysical ($\overline{R_e}_{1/2}$) changes. Optimized asymmetry parameters from

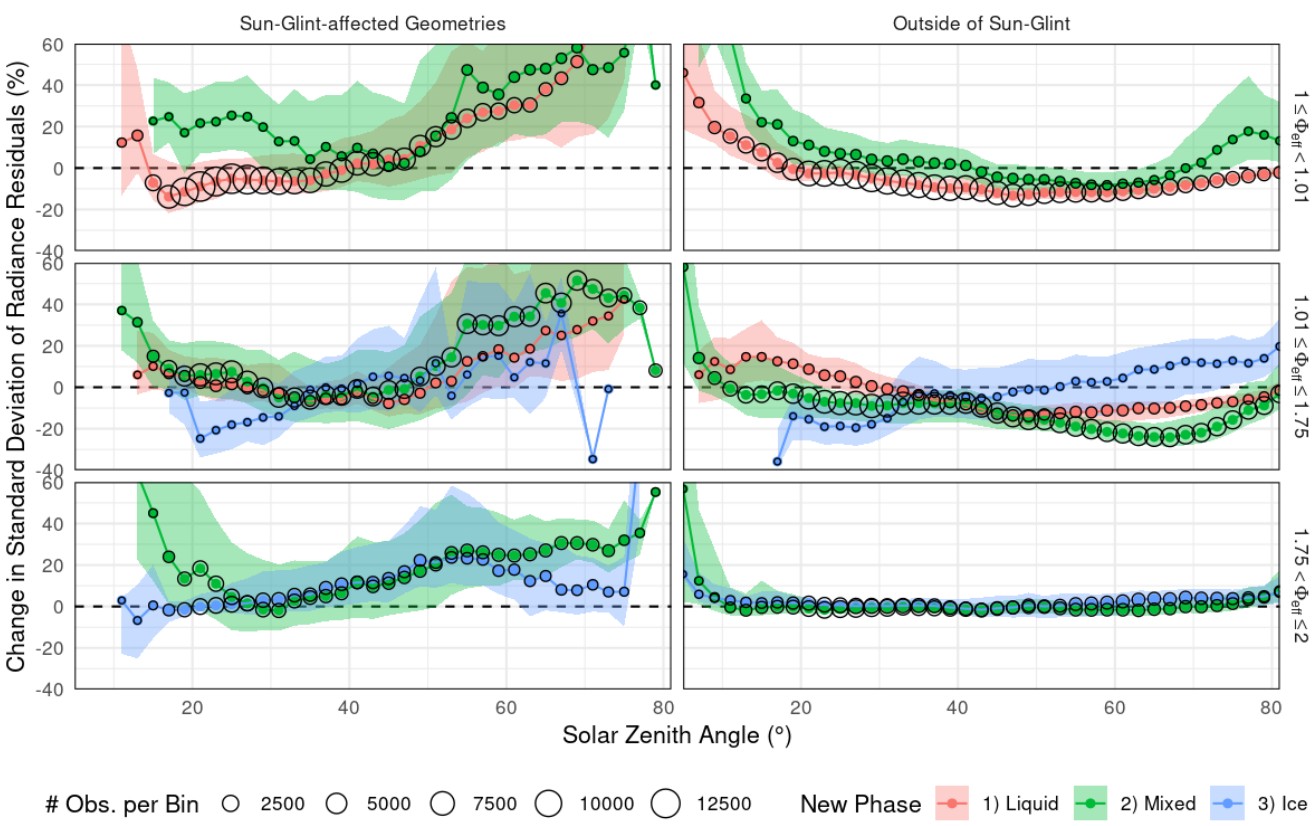

**Figure 4.** Using all observed angular bins within $\theta_0 \in [6°, 82°]$, we show how radiance residuals from proposed log-linear models compare against state-of-the-art sigmoidal fits. Results are presented by CERES-defined cloud phase (vertically), by newly-defined phase (colors), and by whether the angular geometry is affected by sun-glint (left) or free of sun-glint (right). We show relative change in model uncertainty: $\delta = [\sigma(\Delta I^{\text{LogLinear}}) - \sigma(\Delta I^{\text{Sigmoidal}})]/\sigma(\Delta I^{\text{Sigmoidal}}) \cdot 100\%$. Solid lines and dots mark $50^{th}$ percentile and shades show the interquartal range between $25^{th}$ and $75^{th}$ percentiles. Point size relates to the average number of observations per angular bin. The dashed black line marks zero change.

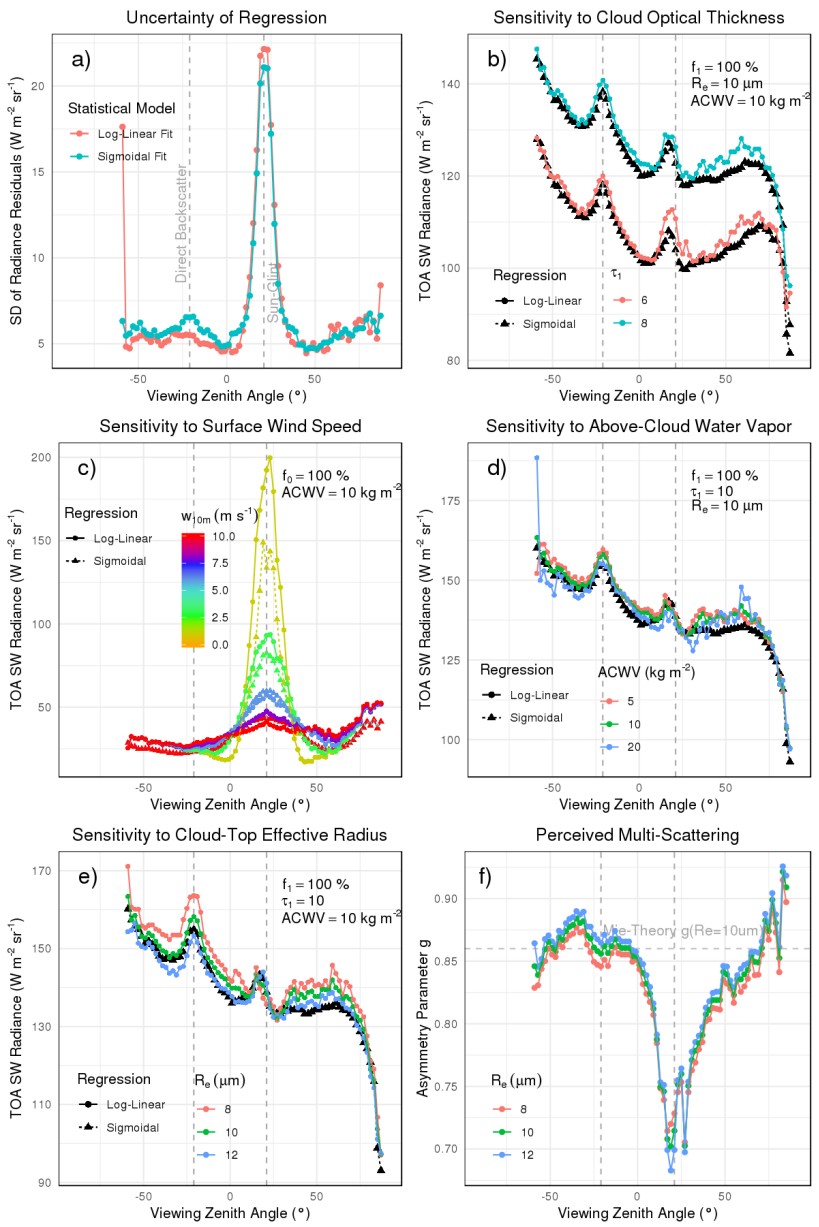

**Figure 5.** For angular bins along the prinicipal plane for $\theta_0 \in [20°, 22°]$ containing liquid-phase footprints, we present error metrics and sensitivities of proposed log-linear versus state-of-the-art sigmoidal fits. (a) shows standard deviations of residuals; colors mark the type of fit. (b) displays the optimal $g(\overline{R_e})$ for three $R_e$ (by color). (c), (d), and (e) demonstrate predicted radiances by both fits for varying cloud optical thickness (c), cloud-top effective radius (d), and above-cloud water vapor (e). Predictions from log-linear fits are colored while predictions from sigmoidal fits are shown in black. (f) presents the response of both fits to a variety of surface wind-speeds. Properties held constant in (c), (d), (e), and (f) are listed in the each panel's top-right corner.





**Figure 6.** For angular bins along the principal plane for $\theta_0 \in [20°, 22°]$, we show details for mixed-phase footprints. (a) presents standard deviations of residuals (colors mark the type of fit). (b) shows optimal $g(\overline{R_e})$ from pure ice and liquid-phase footprints employed for mixed-phase cases (colors mark liquid and ice particle effective radius). (c) and (d) show predicted radiances for liquid cloud fraction (c) and cloud-optical thickness (d). In both (c) and (d), we show log-linear fits in color and sigmoidal fits in black. Quantities left constant are shown in the bottom-left corner.





pure liquid and pure ice cases (Fig. 6b) were reused to describe the cloud albedo of respective cloud phase within each mixed-phase CERES footprint. Hence only $A$, $B$, and $C$ from Eq. 4 needed to be estimated. Fig. 6a illustrates the reduction in model uncertainty for many bins and of up to $2.5\,\mathrm{W\,m^{-2}\,sr^{-1}}$ when using the log-linear approach. Once again, the center of sun-glint remained best captured by the sigmoidal approach. Using a cloud-phase-specific albedo allowed us to account for radiance

changes with varying amount of liquid versus ice fraction within a footprint. Fig. 6c shows radiance predictions for different liquid-ice-proportions (which could not be captured by the state-of-the-art approach) and that both approaches agree for 50% liquid and 50% ice cloud footprints. Fig. 6d shows the sigmoidal fit's sensitivity to ranging cloud optical depth was captured by the log-linear approach. Looking at all available sun-observer-geometries (Fig. 4, middle panels) for solar geometries between $\theta_0 \sim 20°\text{-}70°$, we found model uncertainty of most bins reduced by as far as 35.8%.

In summary, we showed that the proposed log-linear model had the ability to outperform the existing sigmoidal approach in capturing CERES radiance fluctuations per angular bin. It produced lower uncertainties, added new radiance sensitivities, and allowed to treat mixed-phase footprints in a fundamentally different manner.

## 5   Conclusions

Statistical models that capture measurements of TOA SW radiances as a function of corresponding scene type for narrow

sun-observer-geometries are the basis for Angular Distribution Models. In this study, we introduced a new alternative that incorporated additional parameters – namely cloud-top effective radius and above-cloud water vapor – via a semi-physical log-linear approach. We found this new approach to better explain radiance fluctuations for the majority of observed geometries and to produce plausible radiance fields.

Incorporating additional parameters that help explain radiance fluctuations may have minimized sampling bias. Ranges in

effective radius or above-cloud water vapor varied across bins and ignoring this variation can cause a radiance bias in individual angular bins. Even accounting for parameters that may not affect TOA anisotropy, such as cloud horizontal heterogeneity, has the potential to minimize sampling biases. We found varying portions of heterogeneous samples across bins and suspect that their variation in radiance (cf. Fig. 2c) failed to cancel out. Thus, giving higher (or all) weight to homogeneous samples during regression, as done in this study, should eliminate any sampling bias.

The inclusion of cloud-top effective radius and above-cloud water vapor was successful as evidenced by reduced radiance residuals and credible radiance fields. We failed to reduce radiance residuals for ice-phase clouds and made the following observations looking at ice cloud samples. First, among collected observations, we found footprints to mostly contain homogeneous ice clouds. Second, ice clouds had only small loads of water vapor aloft. Lastly, there was an absence of distinct single-scattering features. We suspect that these are characteristics that drive potential reduction of radiance residuals and that

liquid clouds samples, having near asymmetric properties (few homogeneous samples, large loads of water vapor aloft, distinct scattering features), benefitted especially from this new approach.

We successfully used a theoretic framework – inspired by radiative transfer approximations designed for hemispheric averages – and applied it to narrow sun-observer-geometries. A derived byproduct, the asymmetry parameter $g(\overline{R_e}|\theta_0^\Delta, \theta_v^\Delta, \varphi^\Delta)$,





captured observer-specific multi-scattering. Could this byproduct contain information that allows inference on multi-scattering

properties? Monte-Carlo radiative transfer simulations may help in answering this. Future work should simulate radiances, derive simulation-based $g(\overline{R_e}|\theta_0^\Delta, \theta_v^\Delta, \varphi^\Delta)$ and extract additional properties, such as photon path length or number of scattering events.

Statistical models allow finding scene properties that produce similar radiative responses (often referred to as similarity conditions). Like the state-of-art-approach, where different combinations of cloud fraction and cloud optical thickness produced

similar radiances, the new semi-physical approach added cloud particle size and above-cloud absorber mass to parameter combinations. A similarity condition explaining albedo through adjusted optical thickness, $(1-g)\tilde{\tau}$ , was found earlier using simulations (e.g. van de Hulst, 1996). To our knowledge, this is the first time adjusted optical thickness (here employed in the framework of two-stream albedo) has been used to capture similarities of observed radiances.

The proposed semi-physical approach can easily be applied to land surfaces. Imager-based bidirectional reflectance dis-

tribution function (BRDF) products, such as MCD43GF (MODIS BRDF/albedo/nadir BRDF-adjusted reflectance Climate Modeling Grid gap-filled; Moody et al., 2008), could provide land surface albedo and surface bi-directional reflectance in order to determine each observation's footprint albedo. Future efforts should test if this application over land can compete with CERES' separate treatment by latitude-longitude boxes. Recent efforts that demonstrated circumvention of this regional separation for clear-sky ADMs by using MCD43GF instead indicated a positive outcome (Tornow et al., 2019).

Lastly, we hope this new log-linear approach will form the basis of future angular distribution models. In particular, we expect that cloudy scenes of microphysical extremes (i.e. clouds consisting of very small or very large droplets) observed from the backscattering direction will benefit from radiance-to-flux conversion using new models. More accurate estimates of instantaneous fluxes should benefit EarthCARE' studies of cloud-radiative processes regarding both water and energy fluxes. We are currently examining this impact on instantaneous fluxes as well as the propagation of updated flux estimates into daily

and monthly flux products.

*Author contributions.* FT had the idea, designed the experiment, and conducted the research. CD, HB, and RP had major influence on the development of the methodology through discussion. CD and HB further helped revising this manuscript. JF provided essential resources for data processing and evaluation.

*Competing interests.* No competing interests.

*Acknowledgements.* We thank Jason N. S. Cole, Almudena Velazquez Blazquez, Tobias Wehr, and all other members of the CLARA team as well as colleagues at Institute for Space Sciences at Freie Universität Berlin for helpful discussion. We further thank the Atmospheric Sciences Data Center at the National Aeronautics and Space Administration Langley Research Center for providing the Clouds and Earth's Radiant



Energy System Single Scanning Footprint TOA/Surface Fluxes and Clouds data product. We specifically thank Wenying Su for providing us with details on the CERES methodology. This work was possible through funding within the ESA Contract 4000112019/14/NL/CT.



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
