# Peer review of "Using Two-Stream Theory to Capture Fluctuations of Satellite-Perceived TOA SW Radiances Reflected from Clouds over Ocean"

_Atmospheric Measurement Techniques, 2020_

## Referee Comment (RC1) · Anonymous Referee #1 · 15 May 2020

The manuscript of Tornow et al. considers the statistical model which relates the SW radiances and cloud parameters. The paper is suitable for publication in ATM. In principle, it can be published as it is now.

However I suggest some points which can be addressed by authors:

Abstract, line 4: 'Like previous approaches' - sounds confusing was trained -> was statistically trained

Define 'cloud phase' line 78

More general remarks: authors consider very simple radiative transfer models (two-

stream, Eddington etc). Why not to use more accurate models, e.g. MOMO developed by the co-authors?

Several cloud types are considered in the paper. Different cloud types have different expansion coefficients of the phase function. The differences are significant for high order expansion terms. Simplistic radiative transfer models hardly can capture them. I doubt that just asymmetry parameter is sufficient to describe different types of cloud models. In this regard, the choice of the two-stream model needs to be justified. Perhaps, authors could elaborate.
* * *

---

## Author Comment (AC1) · 18 May 2020

We thank the Referee very much for taking the time to review our manuscript. Please find point-by-point responses to the Referee's feedback below.

**Abstract, line 4: 'Like previous approaches' - sounds confusing was trained -> was statistically trained**

We will adopt the proposed changes in the final version.

**Define 'cloud phase' line 78**

We plan to amend "cloud particle phase (involving MODIS band 3.7um)" to "cloud

condensate phase (i.e. liquid, ice, or a mixture of both; involving MODIS band 3.7um)".

**More general remarks: authors consider very simple radiative transfer models (two-stream, Eddington etc). Why not to use more accurate models, e.g. MOMO developed by the co-authors?**

We would like to emphasize that two-stream theory was not used to simulate radiances. Of course, for radiative transfer simulations we would rely on more accurate methods, as the Reviewer suggested.

In this study, we used the functional form of two-stream equations to linearly relate MODIS-retrieved cloud properties with CERES-measured top-of-atmosphere short-wave radiances.

As seen in Fig. 1a, MODIS cloud optical thickness (or more precisely the parameter x, defined in Sec. 3.1) plotted against CERES radiance shows a sigmoidal shape. Because two-stream functions (Eq. 10-12) reproduce this sigmoidal shape, we could link MODIS properties in a linear manner to CERES radiances. Statistical optimization (Sec. 3.2.2) further improved our efforts to explain CERES-measured radiances. The use of higher-order schemes (e.g. four-stream functions) was not tested as two-stream theory produced a satisfactory outcome.

To better emphasize the role of two-stream theory in our manuscript, we plan to make following changes:

- l. 8: instead of "serving as a function of..." put "serving to statistically incorporate..."

- l. 11: instead of "two-stream albedo" put "two-stream functional form"

- l. 58: instead of "to explain cloud albedo based on" put "to statistically ingest"

- l. 154: instead of "two-stream cloud albedo that uses cloud optical thickness and
cloud micro-physical properties" put "two-stream equations as a function to ingest MODIS-retrieved cloud optical thickness and cloud-top effective radius"

**Several cloud types are considered in the paper. Different cloud types have different expansion coefficients of the phase function. The differences are significant for high order expansion terms. Simplistic radiative transfer models hardly can capture them. I doubt that just asymmetry parameter is sufficient to describe different types of cloud models. In this regard, the choice of the two-stream model needs to be justified. Perhaps, authors could elaborate.**

We agree with the Reviewer that the asymmetry parameter (e.g. derived from Mie-theory, and a function of effective radius) is too simple to accurately capture radiance fluctuations for all cloud types and for all viewing-illumination geometries. Because of this limitation, we decided to also statistically optimize the asymmetry parameter (and its change with effective radius). We performed this optimization (Sec. 3.2.2) for each angular bin (i.e. each discrete bin of viewing-illumination geometry) and for each cloud phase (i.e. liquid and ice). For an exemplary angular bin, Fig. 2b presents Mie-calculated asymmetry parameters versus statistically optimized ones.

We hope to have circumvented the limitations of a steady asymmetry parameter by using optimized asymmetry parameters. This allows us to have viewing-illumination-geometry-dependent radiance changes for different effective radii (e.g. effects like the cloud bow and cloud glory). Fig. 5e shows that our approach plausibly produced such radiance changes with effective radius.

To better highlight the use of statistically optimized asymmetry parameters, we plan to make following changes:

- l. 9: after "...footprint." add "Effective radius-dependent asymmetry parameters were obtained empirically and separately for each viewing-illumination geometry."

---

## Referee Comment (RC2) · Anonymous Referee #2 · 17 Jun 2020

**Review of "Using Two-Stream Theory to Capture Fluctuations of Satellite-Perceived TOA SW Radiances Reflected from Clouds over Ocean" by Tornow et al.**

**16 June 2020**

**General comments**

This study applies a semi-physical model to predict the hemispheric radiance distribution of reflected solar radiation by clouds over ocean. When comparing the predicted radiance fields against those from empirically based CERES ADMs, the authors find smaller residuals against CERES observations at many solar-viewing geometries. The value of accounting for cloud effective radius and above cloud water vapor in the new method is highlighted.

The paper is nicely written and of interest to the community. I find it remarkable that a 2-stream cloud albedo and simple representation of radiative transfer can compete with the state-of-the-art CERES ADMs. I do, however, believe that the manuscript would benefit from some more transparency in the relative shortfalls of the new method, and several clarifications. After addressing these minor comments outlined below, I recommend prompt publication in AMT.

**Minor comments**

L13: I find these statistics alone are a bit misleading. After first reading these numbers, I was then slightly disappointed when reading the text and figures to see that some geometries are much worse, and improvements are often much more marginal. Instead of just stating "up to" values, I believe a more honest representation of the results in the abstract should also mention how frequently improvements are seen and/or typical improvements.

L22-23: Is it correct that EarthCARE will use observation based fluxes in the closure assessment? My understanding is that EarthCARE will use observed radiances for this purpose.

L33: CERES ADMs are developed from years of observations, not months. This is actually mentioned later in the manuscript.

L35-36: ERBE only defined 2 scene types containing cloud over ocean. There was also clear sky ocean (technically containing cloud cover up to 5%), and an overcast scene that did not separate surface types. I assume these are the 4 scene types the authors refer to here, but it is probably worth making this distinction.

Eq. 1: Best to define "g" explicitly since it is defined later as the asymmetry parameter. Is there a unit inconsistency in these equations?

L92: Why cut off SZA specifically at 82 deg?

Fig 3: Can you comment on the asymmetry either side of the sun-glint? "Coakley-Chylek refl. surface" gives smaller residuals at viewing zenith angles plotted to the left of the sunglint, but generally worse or comparable to the right. The opposite is true for Fig 6a.

Fig 4: The meaning of the sign of the change should be noted in the caption. I worked out that negative change means the Log-Linear is better, but I had to read the text to get that.

L265-267: Similar to my second comment above about statistics in the abstract, I think these summary sentences over-clam the results somewhat. The proposed log-linear model *sometimes* outperformed the existing sigmoidal approach, but there were also many geometries when it did relatively badly. That should be acknowledged as part of these summary sentences.

*Grammatical corrections*

L13-14: "radiance residuals"->"radiance residuals calculated against CERES observations". It is worth mentioning in the abstract that they are residuals against observations. This may not be obvious to a reader who just picks up the abstract.

L49: Given the importance of water vapor above cloud, I recommend "role of single scattering"->"role of solar absorption and single scattering".

L56: "semi-statistical"->"semi-physical". Better to use consistent language throughout.

L80: "("Note for cloud layer")". I do not understand the meaning of this.

L86: "those" -> "whose"

---

## Author Comment (AC2) · 24 Jun 2020

We thank the Referee very much for providing feedback to our manuscript. Please find point-by-point responses listed below.

**Minor comments**

**L13: I find these statistics alone are a bit misleading. After first reading these numbers, I was then slightly disappointed when reading the text and figures to see that some geometries are much worse, and improvements are often much more marginal. Instead of just stating "up to" values, I believe a more honest**

[Figure]

**representation of the results in the abstract should also mention how frequently improvements are seen and/or typical improvements.**

We thank the Referee for this observation and for proposed solution. We plan on extending the abstract as follows:

l. 12: change "most observer-geometries" to "most observer-geometries outside the sun-glint"

l. 14: "…and 35.8% for footprints containing both cloud phases. Geometries affected by sun-glint (constituting between 10% and 1% of the discretized upward hemisphere for solar zenith angles of 20° and 70°, respectively), however, often showed weaker performance when handled with the new approach and had increased residuals by as much as 60% compared to the state-of-the-art approach. Overall, uncertainties were reduced for liquid-phase and mixed-phase footprints by 5.76% and 10.81%, respectively, while uncertainties for ice-phase footprints increased in uncertainty by 0.34%. Tested for a variety of scenes..."

The results (Section 4) will be amended as follows:

l. 228: "…caused higher uncertainties in log-linear models, increasing with solar zenith angle and higher by up to 60% compared against the sigmoidal approach."

l. 247: "will be discussed in Sec. 5. Similar to liquid-phase clouds, angular geometries affected by sun-glint showed worse performance than the sigmoidal approach, increasing residuals by up to 30%."

l. 259: "…remained best captured by the sigmoidal approach, especially for SZA beyond 50° where the semi-physical model produces up to 55% higher residuals."

And we introduce a summarizing paragraph where we calculate the median change in uncertainty for each of the state-of-the-art-defined cloud phases (i.e. for each of the three rows in Fig. 4):

l. 265: "Across all solar and viewing geometries, we calculated the median change in uncertainty when using the log-linear model over the state-of-the-art approach to be -5.76% for liquid-phase clouds, +0.34% for ice-phase clouds, and -10.81% when both phases are present."

The conclusions (Section 5) will be changed as follows:

l. 273: "...and to produce plausible radiance fields. Weaker performance than the state-of-the-art approach was generally observed for solar zenith angles lower than $20°$ and for sun-glint affected geometries that constitute between 1 and 10% of the hemispheric radiance field."

**L22-23: Is it correct that EarthCARE will use observation based fluxes in the closure assessment? My understanding is that EarthCARE will use observed radiances for this purpose.**

That is correct. The mission objective is defined in terms of TOA fluxes. More specifically, the radiative closure assessment is considered successful when observation-based versus simulated TOA fluxes agree within 10 W/m$^2$ (Illingworth et al., 2015).

**L33: CERES ADMs are developed from years of observations, not months. This is actually mentioned later in the manuscript.**

We will correct this.

**L35-36: ERBE only defined 2 scene types containing cloud over ocean. There was also clear sky ocean (technically containing cloud cover up to 5%), and an overcast scene that did not separate surface types. I assume these are the 4 scene types the authors refer to here, but it is probably worth making this distinction.**

We thank the Referee for this remark and will adapt the text as proposed:

l. 35-36: "and defined four scene types ranging in cloud coverage (including "clear

ocean" that used a cloud cover up to 5%, two cloudy scene types over ocean, and "overcast" that blended all surface types)."

**Eq. 1: Best to define "g" explicitly since it is defined later as the asymmetry parameter. Is there a unit inconsistency in these equations?**

We thank the Referee for pointing this out and will define g explicitly. We double-checked equation 1, found the latter integral lacked a division by pressure p, and will correct this in the final version.

**L92: Why cut off SZA specifically at 82 deg?**

There are several reasons that motivate cutting off before a SZA of $90°$: less reliable MODIS cloud retrievals, a growing influence of twilight, and a progressively smaller influence of cloud micro-physical properties on upward-reflected radiance fields. To demonstrate the feasibility of the log-linear approach as much as possible while keeping computational cost at a minimum, we decided to cut off at $82°$.

**Fig 3: Can you comment on the asymmetry either side of the sun-glint? "Coakley-Chylek refl. Surface" gives smaller residuals at viewing zenith angles plotted to the left of the sunglint, but generally worse or comparable to the right. The opposite is true for Fig 6a**

We thank the Referee for highlighting this difference in performance.

We generally found that liquid-phase clouds (shown in Fig. 3) benefitted from introducing the semi-physical approach, and especially so in the backscattering direction (i.e. left of the sun-glint) where we expect the largest contribution of single-scattering events. We find this confirmed in Fig. 3.

For mixed-phase clouds (shown in Fig. 6), we speculate that a different balance of advantages versus disadvantages of the log-linear model may cause a shift in geometries where the log-linear model outperforms the state-of-the-art approach. Please find these advantages and disadvantages elaborated below.

For mixed-phase clouds (shown in Fig. 6a), we can think of two additional sources of errors that may increase residuals of the log-linear model for any geometry compared to liquid-phase clouds.

First, we used optimized asymmetry parameters from purely liquid-phase and ice-phase footprints for mixed-phase footprints, leaving only 3 parameters to optimize (i.e. A, B, and C from Eq. 4) while the sigmoidal approach used 5 (see Eq. 2) or more in case of sun-glint. We suspect that fewer degrees of freedom could lead to higher residuals in general.

Second, another potential source for larger residuals could arise from determining above-cloud water vapor per footprint from a single cloud-top pressure (Eq. 1). When multiple cloud layers are present we decided to use the cloud-top pressure of the cloud layer with the larger cloud fraction. For mixed-phase footprints - having both ice and liquid phase clouds and, thus, presumably large pressure differences across cloud-tops within each footprint – we expect largest possible uncertainties to arise.

On the other hand, a better performance of the log-linear model may, of course, be found where the intended effect is largest: within a mixed-phase footprint to be able to account for proportions of ice and liquid-phase clouds and, thus, their respective ability to reflect solar radiation (an ability the sigmoidal model loses by producing a footprint-effective cloud optical thickness).

We expect the largest advantage of the log-linear approach for geometries where liquid-to-ice proportions varied the most (or showed most skewed distributions) and presume that this is the case for the forward-scattering direction in Fig. 6a (right of the sun-glint).

Looking at Fig. 6c, we see how predicted radiances from the sigmoidal model can be associated with various ice-to-liquid proportions from the log-linear model along the principal plane: the nadir and forward-scattering direction are associated with ∼75% ice fraction and 25% liquid fraction, while the backscattering direction is associated with 50-50 proportions. This could indicate a shifting distribution in liquid-to-ice proportions between both groups and allow log-linear models to outperform the state-of-the-art

approach in the former group.

As derivatives from this question and its response we plan to make following changes:

l. 91: include "For footprints consisting of multiple cloud layers, relying on a single cloud-top pressure may introduce uncertainty, especially for mixed-phase footprints (see Sec. 3) where the pressure difference between ice and liquid phase layers is exceptionally large."

ll. 261-262: change from "and that both approaches agree for 50% liquid and 50% ice cloud footprints." to "and that both approaches agree for 50% liquid and 50% ice cloud footprints for the backscattering direction and 25% liquid and 75% ice cloud fractions for much of the forward scattering direction, indicating that sampled footprints shifted in liquid-to-ice proportions along the principal plane."

**Fig 4: The meaning of the sign of the change should be noted in the caption. I worked out that negative change means the Log-Linear is better, but I had to read the text to get that.**

We thank the Referee for noting this shortcoming and will expand the caption of Fig. 4 accordingly:

Fig. 4: "…100%. Consequently, negative values relate to a better performance of the log-linear model, while positive values mark a better performance by the state-of-the-art methodology. Solid lines…"

**L265-267: Similar to my second comment above about statistics in the abstract, I think these summary sentences over-clam the results somewhat. The proposed log-linear model sometimes outperformed the existing sigmoidal approach, but there were also many geometries when it did relatively badly. That should be acknowledged as part of these summary sentences.**

We agree with the Referee and - in addition to planned changes listed in the response to the first comment – amend Sec. 4 as follows:

l. 266: Instead of "It produced lower uncertainties", we put "For most geometries it produced lower uncertainties"

l. 267: Adding: "Drawbacks were typically found for geometries affected by sun-glint."

**Grammatical corrections**

**L13-14: "radiance residuals"->"radiance residuals calculated against CERES observations". It is worth mentioning in the abstract that they are residuals against observations. This may not be obvious to a reader who just picks up the abstract.**

We will adjust the text as proposed.

**L49: Given the importance of water vapor above cloud, I recommend "role of single scattering"->"role of solar absorption and single scattering".**

We agree with the Referee and will change the text accordingly.

**L56: "semi-statistical"->"semi-physical". Better to use consistent language throughout.**

We will adapt this as proposed.

**L80: "("Note for cloud layer")". I do not understand the meaning of this.**

We plan to change the text as follows:

l.80: Instead of "("Note for cloud layer")" we put "(using the parameter "Note for cloud layer" from the SSF dataset)"

**L86: "those" -> "whose"**

We will change this.

Lastly, we would like to thank both Referees very much for their feedback and hope to have addressed all questions and comments. To acknowledge their time and effort we

plan to expand the Acknowledgements as follows:

l. 319: "We thank two anonymous referees very much for their feedback that helped to improve this manuscript substantially."

**References**

Illingworth, A. J., and Coauthors, 2015: The EarthCARE Satellite: The Next Step Forward in Global Measurements of Clouds, Aerosols, Precipitation, and Radiation. Bull. Amer. Meteor. Soc., 96, 1311–1332, https://doi.org/10.1175/BAMS-D-12-00227.1.